# Insights of Microstructural Features and Their Effect on Degradation and the In Vitro Bioactivity Response of as-Cast Mg-Sn Alloys for Orthopedic Implant Applications

**DOI:** 10.3390/ma15186327

**Published:** 2022-09-12

**Authors:** Radha Rajendran, Sreekanth Dondapati

**Affiliations:** School of Mechanical Engineering, Vellore Institute of Technology, Chennai 600127, India

**Keywords:** magnesium, tin, corrosion, bioactivity, apatite

## Abstract

The present work focuses on a deep understanding of microstructural evolution and phase formation in a binary Mg-Sn alloy system. Mg-xSn (x = 1, 5, 10 wt.%) alloys were cast using a squeeze casting technique. Phase identification and microstructural analysis were done using XRD (X-ray Diffraction) and FESEM with EDS (Field Emission Scanning Electron Microscopy with Energy Dispersive X-ray Spectroscopy), respectively. The mechanical behavior of the alloys under study was evaluated by conducting a compression test. The corrosion behavior of all the alloys were intricately studied using electrochemical corrosion tests and an immersion test in the simulated body fluid (SBF) environment for different immersion periods. The bioactivity response of Mg-Sn alloys systems under this study was investigated by immersing the samples in SBF for 14 days. From the analysis of the results, it was understood that the amount of Sn addition has a large influence on the metallurgical, corrosion, and bioactivity properties. Interesting facts about the intermetallic phase formation and segregation of Sn were observed when the wt.% of Sn was varied in the alloy and the evolution of the microstructure was described clearly. Mechanical properties of Mg-Sn alloys were improved, as the Sn content increased up to 5 wt.% and declined in the case of a 10 wt.% Sn addition. A similar trend was observed even in the case of corrosion resistance and bioactivity properties. Among the alloy compositions studied, Mg with a 5 wt.% addition has proved to be a promising candidate material for orthopedic implant applications with an acceptable elastic modulus, higher corrosion resistance, and an excellent bioactive response.

## 1. Introduction

Magnesium based materials have received attention in the early 2000s because of their potential properties suitable for orthopedic implant applications [1,2,3]. The density and elastic modulus of Mg and its alloys close to natural bone greatly minimize the stress shielding effect, which are appealing for implant use [4,5]. Mg has good biodegradability, however mismatch with the degradation rate and bone healing period causes adverse effects which hinders the commercialization of Mg as a biomaterial [6]. Hence, the development of a Mg alloy system with a controlled degradation rate that could match the bone healing time, avoiding secondary surgical procedures, will ease the financial burden and pain of patients. Recently, Mg-Sn alloys have been studied extensively for potential application as orthopedic implant material with greater corrosion resistance. Many authors who worked on the Mg-Sn alloy system have concluded that the phases formed in these alloys have a large influence on their corrosion behavior. Palash Poddar et al. investigated the as-cast Mg-8wt%Sn alloys, and they observed that it is comprised of the darker region of an α-Mg matrix, a grey region of eutectic phase consists of α-Mg+Mg_2_Sn, a whiter region of divorced Mg_2_Sn, and tiny Mg_2_Sn particles formed as second constituents [7]. In another study, they observed three different phases, such as primary α-Mg, β-Mg_2_Sn precipitates at the grain boundaries, and a eutectic α-Mg. The α-Mg appeared darker surrounding the β-Mg_2_Sn precipitates and had irregular geometry, which is divorced in nature [8]. Ozgun et al. fabricated the Mg-Sn alloys by varying Sn content (5 wt.%, 9 wt.%, and 13 wt.%) by powder metallurgy, and they observed the formation of two phases (i) α-Mg and (ii) Mg_2_Sn at the grain boundaries. The Mg_2_Sn phase was identified as the whiter region in their study, however, the same could not be confirmed by their EDS elemental mapping results [9]. Pingli Jiang et al. analyzed the microstructure of Mg-Zn-Sn alloys, and they found the presence of different morphological features of the phases, which appeared as irregularly oriented divorced eutectic phases. The brighter contrast region showed the enrichment of Sn and Zn microconstituents [10]. Xuejian Wang et al. stated that the eutectic Mg_2_Sn phase appeared as the brighter region and the greyish bands show the enrichment of Sn along the grain boundaries in Mg-5wt%Sn alloy [11]. Zhen et al. concluded that the Mg-3Sn-Mn alloy composed of hard Mg_2_Sn precipitates in the primary Mg matrix could help to improve its strength. Further, they have also stated the addition of Sn helps to stabilize the formation of hydroxide layers, which prevents acceleration of the corrosion process [12]. Kubasek et al. investigated the morphology of as cast Mg-Sn alloys with various Sn content (1 wt.%, 5 wt.%, and 7 wt.%) and observed that it consists of α-Mg and Mg_2_Sn intermetallic particles at the interdendritic regions. The volume of hard phases increases with an increase in Sn content, and it is highly beneficial for improving its strength and hardness. It was also mentioned that Sn alloying helps to improve corrosion resistance by forming SnO_2_, which is more stable in a larger range of pH, thus keeping the Mg(OH)_2_ layer intact on the surface [13]. Zhou et al. analyzed the microstructure of laser melted Mg-Sn alloys and observed that it consists of primary Mg, the Mg_2_Sn phase, and elemental Sn. The brighter region in the micrographs was observed as the enrichment of Sn whereas the eutectic Mg_2_Sn phases have appeared as a semi continuous network in the Mg matrix [14]. Zhao et al. fabricated the Mg-xSn (1wt.%, 3 wt.%, 5 wt.%, and 7 wt.%) alloys by laser melting and observed that it comprised of α-Mg and Sn rich phase segregation. They analyzed the microstructure and claimed that enrichment of Sn occurred besides the formation of the Mg_2_Sn intermetallic phase, owing to the faster solidification rate in salt water [15]. Liu et al. observed the morphology of Mg-Sn alloys, which is composed of primary Mg matrix, eutectic Mg_2_Sn, devoiced Mg_2_Sn, and Mg_2_Sn particles as second constituents [16]. Mg-Sn alloys with a higher weight percentage of Sn (24, 37, and 50%) were also fabricated, and it was found that the large amount of eutectic phase (Mg_2_Sn) was formed with a higher weight percentage of Sn, along with the formation of α-Mg and Mg_2_Sn particles [17]. The addition of Sn in the Mg matrix acts as a barrier for corrosion reaction by forming SnO_2_. It keeps the protective layer intact and prevents the inhibition of Cl^−^ ions thereby improving corrosion resistance. The hydrogen evolution rate was observed less in Mg-5Sn alloy and corrosion pits were also observed to be less compared to pure Mg. The enrichment of Sn was also helpful for reducing the H_2_ evolution [18,19,20]. In another study, it was also mentioned that Sn has a favorable effect on both castability and corrosion resistance, even in Mg-Sn alloy composites [21,22].

From the above literature, it can be understood that there is no clear understanding of the phases formed and their mechanism to prevent corrosion in the Mg-Sn alloy system. Hence, the present research focuses on a thorough analysis of microstructural evolution in Mg-Sn alloys and the effect of Sn on corrosion behavior.

## 2. Materials and Methods

### 2.1. Fabrication of Mg-Sn Alloys

Mg-xSn (1 wt.%, 5 wt.% and 10 wt.%) alloys were fabricated by a squeeze casting technique. Pure Mg and Sn ingots of 99% purity were charged in the preheated crucible, which is operated by a microprocessor-based electrical resistance furnace under controlled inert argon gas atmospheric conditions. The molten charge was maintained at a temperature of 750 °C and thoroughly stirred by using a twin blade stainless steel stirrer operated at 400 rpm for 10 min. Then, the molten Mg-Sn alloy was transferred to the preheated mild steel mould through the inclined hopper at the bottom of the furnace. A 40-ton hydraulic press was then operated to plunge the piston for squeezing the molten metal at the pressure of 150 MPa for 60 s. The rapid solidification of the melt occurs through the application of pressure and the cast Mg-Sn sample is ejected.

### 2.2. Phase Identification and Microstructure

The phases present in the samples under this study were evaluated by analyzing the X-ray Diffraction (XRD) spectrum obtained by X-ray diffractometer (X’pert Pro, PANalytical, Almelo, The Netherlands) with Cu K*α* radiation (*λ* = 0.1540598 nm) at 45 kV and 30 mA. The scan speed of 1°/min and a step size of 0.05*°* over the 2θ range of 20° to 90° were employed during the scanning. Field Emission Scanning Electron Microscopy (FESEM) equipped with Energy Dispersive Spectroscopy (EDS) (EVO/18, Zeiss, Jena, Germany) was used to observe the surface morphologies of as cast composites and also to detect the presence of elements in all the samples.

### 2.3. Mechanical Characterization

The cylindrical specimens with L/D ratio ~1 were machined from the cast billets and subjected to compression tests as per ASTM E9-89A, using Schimadzu servo con- trolled machine with a cross head speed of 0.154 mm/min. The mechanical properties such as Ultimate Compressive Strength (UCS), Yield Strength (YS), and elastic modulus for all the samples were obtained. The tests were repeated five times for each composite and the average values of results are presented in Table 1.

### 2.4. Electrochemical Corrosion Measurements

The Tafel and Electrochemical Impedance Spectroscopy (EIS) tests were performed in SBF medium soaked for 1 h, 24 h, 48 h and 72 h at 36.5 °C using a potentiostat (Inter- face 1010, Gamry Instruments, Warminster, PA, USA) by exposing 0.375 cm^2^ area of the sample, respectively. The SBF was prepared as per the composition of materials proposed by kokubo [23], such as NaCl (8.035 g), NaHCO_3_ (0.355 g), KCl (0.225 g), K_2_HPO_4_.3H_2_O (0.231 g), MgCl_2_.6H_2_O (0.311 g), CaCl_2_ (0.292 g), Na_2_SO_4_ (0.072 g), Distilled water (1000 mL), HCl, Tris, and the pH of the solution was maintained at 7.45. The samples for the electrochemical test were polished to 1200 grit size and cleaned with ethanol before testing to ensure homogeneity of the surface. The corrosion cell entailed a Saturated Calomel Electrode (SCE) and a platinum wire, as reference and counter electrodes, respectively. Tafel plots were generated by polarizing the specimen approximately 0.3 V anodically and cathodically, with reference to Open Circuit Potential (OCP) at a scan rate of 0.5 mV s^−1^ after an initial delay of 60 min. EIS measurements were made from a start frequency of 104 Hz up to 0.1 Hz with an AC signal amplitude of 10 mV rms. After the measurements were obtained, the Tafel and impedance data were analyzed by curve fitting and equivalent circuit modeling using Gamry Echem Analyst software (1010T).

### 2.5. Immersion Test

The samples for the immersion test were polished to 1200 grit size and cleaned with ethanol before the immersion test was performed. The immersion test was performed in SBF for 24 h, 48 h and 72 h at 36.5 °C in a constant temperature bath. The ratio of SBF to the surface area of the sample was 100 mL/cm^2^ to minimize the change in pH of the SBF during the test. The periodic measurements of the pH of the solution were carried out to monitor the degradation process directly related to the release of OH^−1^ concentrations. Thus, a lower pH can be interpreted to a better corrosion resistance profile of the sample. The SBF was reloaded after immersion testing conditions. The samples were weighed before and after each immersion time; the samples were thoroughly rinsed and subjected to ultra-sonication in ethanol and also in the mixture of 180 g/L chromic acid and 10 g/L silver nitrate to remove the corrosion products according to ASTM G31. The corrosion rate from the weight loss measurements was calculated using Equation (1),
C.R = K.w/d.A.t(1)
where:C.R is the corrosion rate in mpy;K is the constant taken as 3.45 × 10^−6^;w is the weight loss in grams;d is the density of the sample in g/cm^3^;A is the surface area of the samples in cm^2^;t is the time in hours.

### 2.6. In Vitro Bioactivity Test

For the in vitro bioactivity test, circular discs of samples were machined to the dimensions of 10 mm in diameter and 5 mm in thickness from the cast billets. The surface area of all the samples was ensured to be 314.16 mm^2^. All the samples under the study are immersed in 31.4 mL of SBF and kept inside a constant temperature water bath maintained at 36.5 °C for 14 days. At the end of the immersion period, samples were taken out, rinsed thoroughly with distilled water, and dried in a desiccator for 2 days. The bioactivity of the samples was characterized for the apatite layer [24,25,26,27] formation observed on the surface of the samples from their FESEM micrographs.

## 3. Results and Discussion

### 3.1. Phase Analysis

XRD patterns of Mg-Sn alloys under this study are shown in Figure 1. The major peak of Mg_2_Sn was identified at 2θ = 22.783°, according to the Joint Committee on Powder Diffraction Standards (JCPDS) card number 06-0190 in Mg-5Sn and Mg-10Sn. It can also be observed that the Mg_2_Sn phase content increased with an increase in the content of tin. Hence, it can be inferred that the greater the tin content, the greater the Mg_2_Sn phase. No other phases corresponding to Mg and Sn were identified. All the other peaks were corresponding to *α*-Mg, which was indexed according to the JCPDS 01-1141. In the case of Mg-1Sn, the peak could not be identified probably due to the very low phase content. This can also be confirmed from the phase diagram of the Mg-Sn binary alloy system shown in [28]. According to the phase diagram, the solubility of Sn in Mg is relatively high, with 14.85 wt.% at the temperature of 561.2 °C. It decreases sharply and reaches almost zero at room temperature. The large substantial solubility limit of Sn in Mg helps tin atoms easily precipitate and form Mg_2_Sn intermetallic secondary phases.

### 3.2. Microstructural Analysis

Figure 2a–c illustrates the scanning electron micrographs of Mg-1Sn, Mg-5Sn, and Mg-10Sn, respectively. All Mg-Sn alloys showed the dendritic microstructure of α-Mg matrix with the Mg_2_Sn eutectic phase (α-Mg+Mg_2_Sn) at the arms of the dendrites and a brightly lit Sn rich area around the grain boundaries. Mg-1Sn alloy consists of the Mg_2_Sn eutectic phase and bright particles of the Sn rich phase around the dendritic boundaries. The quantity of the Mg_2_Sn phase seems to be higher as compared to the Sn rich phase. This can be confirmed from the elemental mapping of the Mg-1Sn alloy shown in Figure 3a–c. It can be observed from Figure 3b that in the area where the Mg element was absent, the intensity of the Sn element was found to be higher, as can be seen in Figure 3c, which confirms that it could be an Sn rich area. Further, the presence of the Sn element all over the matrix (Figure 3c) indicates the dissolved tin in α-Mg. As the tin content increased to 5 wt.%, the eutectic Mg_2_Sn phase content as well as the Sn rich phase seems to have increased and formed along the cell boundaries by macro segregation during solidification, as can be seen from Figure 2b. In contrast to Mg-1Sn, the Sn rich phase has formed a semi-continuous network in Mg-5Sn. Further, the microstructure of Mg-5Sn showed the cellular structure rather than the dendritic structure, as in Mg-1Sn alloy. The appearance of the Sn-rich phase has also changed from particulate to semi-continuous along the cellular boundaries, which can also be confirmed from the EDS elemental mapping results depicted in Figure 3d–f. With a further increase in tin content to 10 wt.%, the Sn rich phase was more prominently present along the grain boundaries and observed to be greater in quantity as compared to both Mg-1Sn and Mg-5Sn alloys. However, with the increase in tin content, the Mg_2_Sn phase also increases as already seen from the XRD results as well as the EDS elemental mapping of Mg-10Sn alloys, as shown in Figure 3g–i.

Two major observations can be made from the microstructures of Mg-1Sn, Mg-5Sn, and Mg-10Sn alloys. Firstly, the phase quantity of both Mg_2_Sn and Sn rich area increases with the increase in Sn content in the alloy. Secondly, with the increase in Sn content, significant microstructural refinement occurs. The formation of the Sn rich phase can be explained with the help of constitutional supercooling (CS) that occurs during the solidification process. As the α-Mg dendrite grows into the solution, the liquid solution ahead of the solid phase will be enriched with Sn, which then transforms to Mg_2_Sn. Further, the rapid cooling rates achieved by the squeeze casting process aids in the segregation of the Sn rich phase along with the Mg_2_Sn phase. The amount of CS increases generally with the concentration of the alloying element. Similarly, when the amount of Sn increases in the alloy, constitutional supercooling will also be higher, which ejects more amounts of Sn between dendritic arms. The Sn thus formed will not mix in the remaining liquid due to the structure of the dendrite and partitioned over the length of secondary dendrite arm spacing. This higher amount of tin and Mg_2_Sn phases restrict the dendritic grain growth and hence grain refinement occurs. This is the reason why Mg-1Sn has coarser dendrites and when tin content is increased to 5 wt.%, coarser dendrites become finer. With the further increase in tin content to 10 wt.%, significant grain refinement was observed. Ideally, the microstructure of Mg-Sn alloys should contain α-Mg and Mg_2_Sn phases at room temperature in accordance with the Mg-Sn phase diagram. The deviation in the microstructure of Mg-Sn alloys observed in this study can be attributed to the squeeze casting process, which uses faster cooling rates.

### 3.3. Mechanical Properties

The stress–strain curve of Mg-Sn alloys with various Sn concentrations is shown in Figure 4. The compression strength of Mg-Sn alloys was found to be 246.86 MPa with 1 wt.% Sn and increased to 271.39 MPa with 5 wt.% Sn.

However, with a further increase in the Sn concentration to 10 wt.%, the compression strength declined to 244.78 MPa. A similar trend was observed for elastic modulus and yield strength except for % elongation, as seen in the Table 1. The % elongation seems to have decreased with an increase in the content of Sn. The increase in strength and decrease in elongation can be attributed to the grain refinement and an increase in the volume of the Mg_2_Sn phase when tin content increased from 1 wt.% to 5 wt.%. Both grain refinement and the Mg_2_Sn phase along the grain boundaries effectively arrest the dislocation movement thereby increasing the strength. However, the same trend could not be observed when Sn content was increased to 10 wt.%. The reason for this behavior can be ascribed to the enormous amount of Sn presence along the grain boundaries when compared to the Mg_2_Sn phase, as observed from Figure 2c.

Hence, it can be understood that the presence of a hard intermetallic Mg_2_Sn phase, which has a high melting point favors the improvement in strength. Whereas, the presence of a low melting point Sn rich phase at the grain boundaries in Mg-10Sn alloy deteriorates the mechanical properties and can prove detrimental. Furthermore, the elastic modulus also increased with an increase in Sn content from 1 wt.% to 5 wt.% and decreased when Sn content was increased to 10 wt.%. As seen in Table 1, the elastic modulus values are closer to that of bone, which can help in reducing the stress shielding effect when used as an orthopedic implant material. It can be clearly understood that the increase in Sn content beyond 5 wt.% is detrimental as it deteriorates the mechanical properties.

### 3.4. Tafel Analysis

The results of the Tafel test for Mg-Sn alloys after exposure in the SBF medium for a short time, 24 h, 48 h, and 72 h are shown in Figure 5, a through d, respectively, and tabulated in Table 2.

It can be observed from the corrosion current density (i*_corr_*) values that the corrosion resistance has shown an increasing trend when Sn content increased from 1 wt.% to 5 wt.% in Mg-Sn alloys. Upon increasing Sn content to 10 wt.%, the corrosion resistance declined.

It can be inferred that more than 5 wt.% of Sn could be detrimental in terms of corrosion resistance. This can be attributed to the presence of Sn enrichment in the alloy which results in galvanic couple formation between Mg and Sn. As the standard electrode potential difference between Mg (−2.37 V) and Sn (+0.15 V) is large, they tend to undergo galvanic corrosion. All Mg-Sn alloy systems under this study have shown the trend of increase in corrosion resistance as a function of immersion time, which is in agreement with the pH of the SBF solution measured during the different immersion periods, as shown in the Table 3. The decrease in the pH of the solution with the immersion time can be explained as follows: Initially, when the SBF solution comes in contact with the Mg surface, the dissolution of Mg occurs by the following reactions:Mg → Mg^2+^ + 2 e^−^
H_2_O → 2 H^+^ + OH^−^

The free electrons lost by Mg will react with H^+^ ions in the solution and become H_2_ gas. The OH^−^ ions dissociated from H_2_O will react with Mg^2+^ ions and form Mg(OH)_2_ by the reaction:Mg^2+^ + 2 OH^−^ → Mg(OH)_2_

As this reaction requires the consumption of OH^−^ ions from the solution, the concentration of these ions in the solution will decrease and hence the pH. In the present study, it is observed that with an increase in the immersion period, the pH of the solution decreased continuously. This behavior could be because of the growth of the Mg(OH)_2_ layer as a function of immersion time. As the Mg(OH)_2_ layer grows in thickness, the probability of production of Mg^2+^ ions reduces. When Mg^2+^ ions production has lessened, the availability of free electrons for the formation of H_2_ gas largely reduces, which in turn increases the H^+^ ions concentration in the solution. Owing to this, the pH of the solution will tend to decrease. Thus, the decrease in the pH of the SBF solution indicates the formation of a thicker and non-porous Mg(OH)_2_ layer on the substrate, which in turn helps to provide better corrosion resistance. Furthermore, since the corrosion medium is SBF in this case, the deposition of calcium phosphates on the top of the Mg(OH)_2_ layer can be expected, which also helps to seal off the pores or cracks, if any, in the Mg(OH)_2_ layer. In Mg-Sn alloys, Sn plays a major role in improving corrosion resistance. During the corrosion process, Sn forms SnH_2_ by reacting with H_2_O initially. Later, it is converted to SnO_2_ as per the E-pH diagram of Sn-H_2_O [29]. This SnO_2_ layer which forms on the surface of the substrate underneath the Mg(OH)_2_ layer plays a major role in protecting the alloy from Cl^−^ ions attack, as compared to the Mg(OH)_2_ layer. It is interesting to note that the rise in pH of the SBF solution for Mg-1Sn and Mg-10Sn is as high as 10.61 and 10.30, respectively, after 24 h of immersion and a similar trend was observed even for 48 h and 72 h of immersion. This indicates the high concentration of OH^−^ ions in the solution due to the faster rate of degradation of Mg, which results in a larger amount of hydrogen evolution. It can also be observed that the rise in the pH of the SBF solution in the case of the Mg-5Sn alloy was not as high as that observed in the Mg-1Sn and Mg-10 Sn alloy system which in turn indicates the slower rate of degradation of the alloy. Furthermore, with the increase in immersion time, the pH of the SBF solution decreased constantly in the case of Mg-1Sn and Mg-5Sn alloys, whereas in the case of Mg-10Sn the pH started to increase again after 48 h of immersion. This infers that the hydroxide layer that was formed on the surface of the Mg-10Sn alloy was not stable after 48h of immersion, which can be attributed to the domination of galvanic corrosion over the protection offered by the SnO_2_ film. In addition, the Cl^−^ ions in the solution also start to diffuse into the Mg(OH)_2_ layer through the cracks and further accelerate the corrosion by the following reaction:Mg^2+^ + 2 Cl^−^ → MgCl_2_

A little observation of the SEM micrographs of Mg-1Sn, Mg-5Sn, and Mg-10Sn alloys also reveals the reason why the corrosion resistance has increased up to 5wt.% of Sn and decreased. It can be observed that the segregation of Sn was found to be less in the case of Mg-5Sn alloy, which increased drastically in the Mg-10Sn alloy. Hence, it can be hypothesized that the corrosion protection in the case of the Mg-1Sn and Mg-5Sn alloy system was mainly due to the SnO_2_ surface film formation, whereas the significant decrease in corrosion resistance of Mg-10Sn alloys was attributed to the formation of the galvanic couple between Mg and Sn since the larger amount of Sn segregation, which was present in Mg-10Sn alloys, accelerates the galvanic corrosion incapacitating the protection offered by SnO_2_ film.

### 3.5. EIS Analysis

EIS is an effective technique for studying the electrochemical corrosion process on metals. It is characterized by various mechanisms, including mass transfer, charge transfer, and diffusion. EIS analysis provides information on the corrosion mechanism. A comparison of the impedance data obtained for Mg-Sn alloys under a different immersion period is shown in Figure 6a–d.

The electrochemical parameters obtained by fitting the EIS curves using an equivalent electrical circuit shown in Figure 7 are tabulated in Table 4. In the equivalent circuit, charge transfer resistance (R*_ct_*) represents the charge transfer resistance; outer layer resistance (R*_o_*) represents the inner layer resistance, constant phase element corresponding to charge transfer resistance (Y*_ct_*), and constant phase elements corresponding to R*_ct_* and R*_o_*, respectively. The solution resistance is given by solution resistance (R_s_) and an inductance L is used in series with the resistance R_L_. All the Nyquist plots have shown two capacitive loops, one at the high frequency and the other at the medium frequency, along with an inductive loop in the low frequency region. Generally, the high and medium frequency characteristics are associated with electrolyte penetration and intrusion, whereas the low-frequency region conveys information on the electrode control process as well as the contribution from localized defects to the overall impedance. The behavior of the loop at high, medium, and low frequencies may be attributed to changes in the corrosion product layer and mode of corrosion. In the Nyquist curves of a short time immersion period, the two capacitive loops at a high and medium frequency range and a well-defined inductive loop at a low frequency range can be observed. This is due to the formation of an oxide layer soon after immersion in SBF. The capacitive loops indicate that the corrosion mechanism was mainly due to the charge transfer process. The diameter of the capacitive loop in the case of the Mg-5Sn alloy was found to be larger, followed by Mg-1Sn and Mg-10Sn alloys. The polarization resistance can be realized by adding R*_ct_* and R*_o_* for all the alloy systems. The charge transfer resistance and the inner layer resistance of the Mg-5Sn was found to be higher, indicating a higher corrosion resistance. The inductive behavior in the case of the short term immersion period for all the alloys is attributed to non-uniform corrosion and the relaxation of adsorbed species, such as Mg(OH)_2_. In the case of the 24 h immersion period also, Mg-5Sn showed better corrosion resistance as compared to the other two alloy systems. However, unlike in the short term immersion period, the Mg-10Sn alloy showed a better corrosion resistance compared to Mg-1Sn. This could be due to the quick formation of a stable SnO_2_ layer on the surface, which is evident from the R*_o_* value, which is higher than that of other alloys. However, after 48h of the immersion period, the behavior of Mg-10Sn has changed again and shown poor corrosion resistance as compared to other alloys. This can be ascribed to the onset of galvanic corrosion due to the galvanic coupling between α-Mg and β-Sn. The reason for Mg-1Sn to show higher corrosion resistance than Mg-5Sn was unpredictable. After 72 h of the immersion period, the Nyquist curve showed more organized capacitive loops owing to the thickening of the oxide layer and improved protection thereof. Furthermore, the inductive loops were almost invisible, which can be explained as the corrosion product layer was more stable. Hence, it can be clearly understood that Mg-5Sn shows excellent corrosion resistance after 72 h of immersion due to the strong protection offered by the SnO_2_ film on the surface underneath the stable Mg(OH)_2_ layer, which was identified by XRD, as depicted in Figure 8. Ideally, with the higher content of 10 wt.% Sn, the Mg-10Sn alloys should have shown better corrosion resistance because of an ability to more quickly form an SnO_2_ film. However, galvanic corrosion has taken over the beneficial effects of the SnO_2_ film, leading to lower corrosion resistance. The bode plot has advantages over the Nyquist plot. Since frequency appears as one of the axes, it is easy to understand from the plot how the impedance depends on the frequency. The phase angle and impedance curves of bode plots of Mg-Sn alloys under different immersion periods are shown in Figure 9a–d (a1–d1), respectively. The phase angle curves of all the immersion periods show a single-phase maximum in the medium frequency range. The higher phase angle corresponds to the higher corrosion resistance. The trend of polarization resistance behavior of all the alloys during all the immersion periods were observed to be similar to that previously explained with the help of Nyquist plots. However, the phase angle curves show shift of phase maxima on the axis of frequency. The phase maxima shift towards the higher frequencies indicate the faster rate of reactions in electrochemistry in general. The fastest process is the combination of corrosion product layer resistance with double layer capacitance, which appears at the higher frequency. It can be observed from Figure 9a–d that the phase angle curve of Mg-10Sn alloy exhibited a peak shift toward the higher frequency for all the immersion periods except short time immersion. This implies that Mg-10Sn alloy has shown a faster corrosion rate as confirmed from the Tafel results, as seen in Table 2. Furthermore, the increase in the phase angle indicates that the oxide layer is becoming more porous and feebler, although this layer’s thickness is increasing at the same time. As observed from Figure 8, the value of the maximum phase angle (θ), increases from −42° to −25° in the case of the Mg−1Sn alloy, −48° to −45° for the Mg−5Sn alloy, and −35° to −20° for the Mg-10Sn alloy, with the increase in immersion period from a short time to 72 h. It is worth noting that the Mg-5Sn alloy showed less of an increase in phase angle from a short time to the immersion period of 72 h, which clearly manifests that the oxide layer is less porous and more intact, hence showing improved corrosion resistance as compared to that of other alloys. The Z curves of bode plots in Figure 9 (a1–d1) depict the polarization resistance along with ohmic resistance at lower frequencies read from its horizontal plateau. At the mid frequency range, the Z curves should ideally show a 45° line with a slope of −1. In the present study the curves are not showing such behavior owing to the non-uniform corrosion kinetics. However, the Z curves of the Mg-5Sn alloy shows the behavior close to that of the ideal curve, which clearly indicates the uniform passivation on the surface. At the highest frequency range, all of the curves exhibited solution resistance, values of which are closer to each other, as can be observed from Figure 9 (a1–d1).

### 3.6. Bioactivity Response

The surface morphologies of Mg-Sn alloys after exposure in SBF for 14 days are studied using SEM analysis and are depicted in Figure 10. In general, degradation occurs rapidly across the surface of the Mg alloys during the initial immersion period. After immersion of all the Mg-Sn alloy samples for 14 days in SBF, it is observed that the surface of all the samples are covered with the layer majorly consisting of Mg(OH)_2_ and Ca and P deposition spread across the entire surface. To be more precise, the Ca and P deposition is more in the case of the Mg-1Sn and Mg-5Sn samples as compared to Mg-10Sn, which is evident from Figure 10. It can be clearly observed that the surface of Mg-1Sn and Mg-5Sn was covered with the spherical shaped deposits (white colored in Figure 10), whereas the surface of Mg-10Sn was covered with a flowery, structured deposition. This can be clarified more from the EDS elemental mapping results, as shown in Figure 11. It is very evident that the presence of the elements Ca and P in the Mg-10Sn sample is very low, whereas Mg-5Sn has a maximum amount of Ca and P presence uniformly spread across the surface. The presence of more amounts of Mg and O in the Mg-10Sn sample clearly demonstrates that the formation of Mg(OH)_2_ is higher than the deposition of Ca and P.

Hence, it can be understood that the bioactivity performance of the Mg-Sn alloys in this study follows the same trend as that of the corrosion behavior. Obviously, the corrosion rate of the Mg-10Sn would have caused the loosening of Ca and P layers even if they have formed. 

## 4. Conclusions

In the present study, Mg-1wt%Sn, Mg-5wt%Sn, and Mg-10 wt%Sn alloys were meticulously investigated for their microstructural features, mechanical properties, corrosion behavior, and bioactivity performance. From this study, it can be concluded that the percentage of Sn more than 5 wt.% could be detrimental to the mechanical, corrosive, and bioactive properties. Up to a 5 wt.% addition of Sn to Mg would improve the mechanical, corrosion, and bioactive properties, thereafter they start deteriorating. The important findings drawn from this study are as follows:(i).The major phases found in Mg-Sn alloys are α-Mg (dark grey), Mg_2_Sn phase (light grey), and Sn rich phase (brightly lit);(ii).The segregation of Sn occurred due to the constitutional super cooling effect due to rapid solidification aided by the squeeze casting process;(iii).The segregation of Sn increases with the increase in wt.% of Sn in Mg alloy;(iv).There was an improvement of ~58% in yield strength when Sn content increased from 1 wt.% to 5 wt.%. However, there was an approximately equivalent percentage of decrement in yield strength when Sn content further increased to 10 wt.%;(v).From the results of electrochemical corrosion and immersion tests, it was observed that the corrosion rate of Mg-5wt%Sn alloy decreased by ~40%, whereas there was an increase of ~50% in the corrosion rate when the addition of Sn was increased to 10 wt.%;(vi).It was concluded that the higher corrosion resistance of Mg-5wt%Sn was due to SnO_2_ film formation, which aids in the formation of apatite layer thereby reducing the release of OH^−^ ions from the surface as evident in the pH measurements. Furthermore, the pH value gradually decreased from 9.6 to 8.34 with the immersion time, indicating the effectiveness of SnO_2_ film, which is not the case for Mg-1wt%Sn and Mg-10wt%10Sn alloys;(vii).The bioactivity response of all Mg-Sn alloy compositions was in line with corrosion behavior. The calcium and phosphate deposition were higher on the surface of Mg-5wt.%Sn alloy as compared to Mg-1wt.%Sn and Mg-10wt.%10Sn alloys.

## Figures and Tables

**Figure 1 materials-15-06327-f001:**
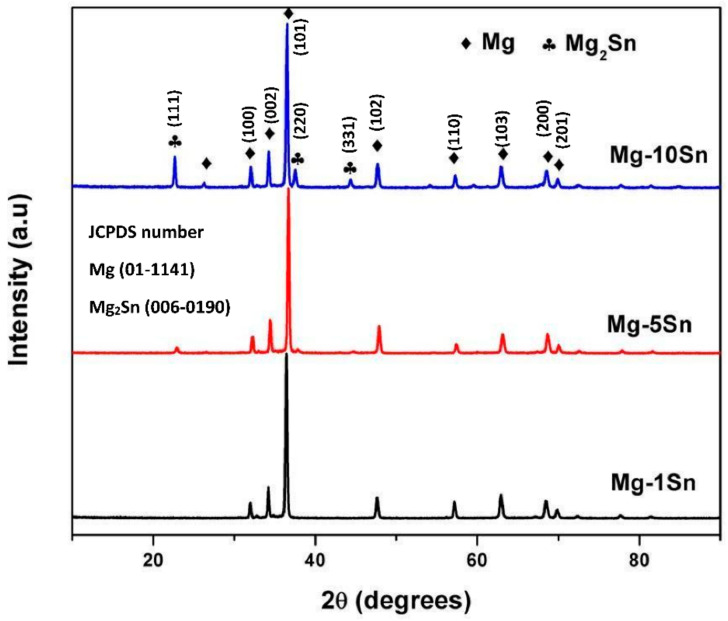
XRD spectra of Mg-Sn alloys.

**Figure 2 materials-15-06327-f002:**
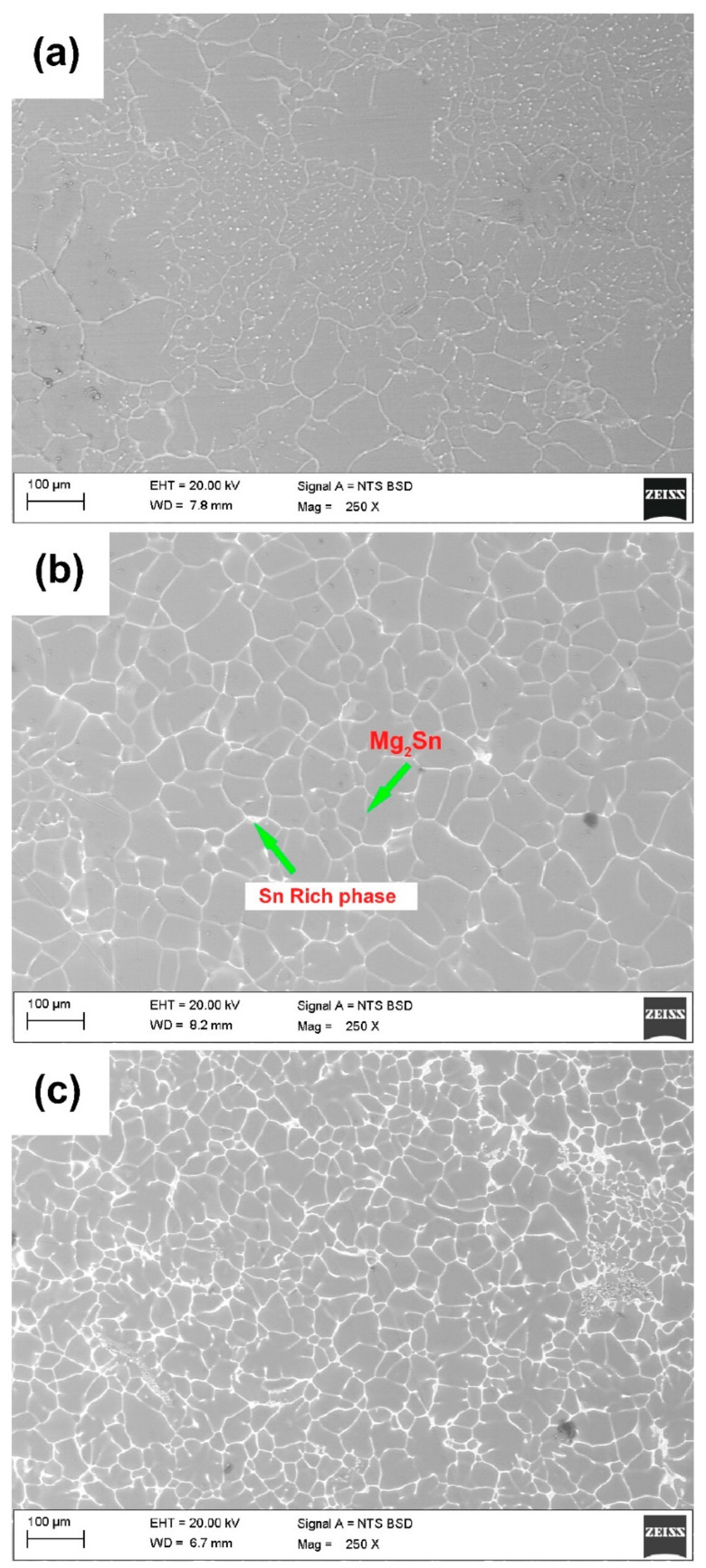
FESEM micrographs of (**a**) Mg-1Sn; (**b**) Mg-5Sn; (**c**) Mg-10Sn alloys.

**Figure 3 materials-15-06327-f003:**
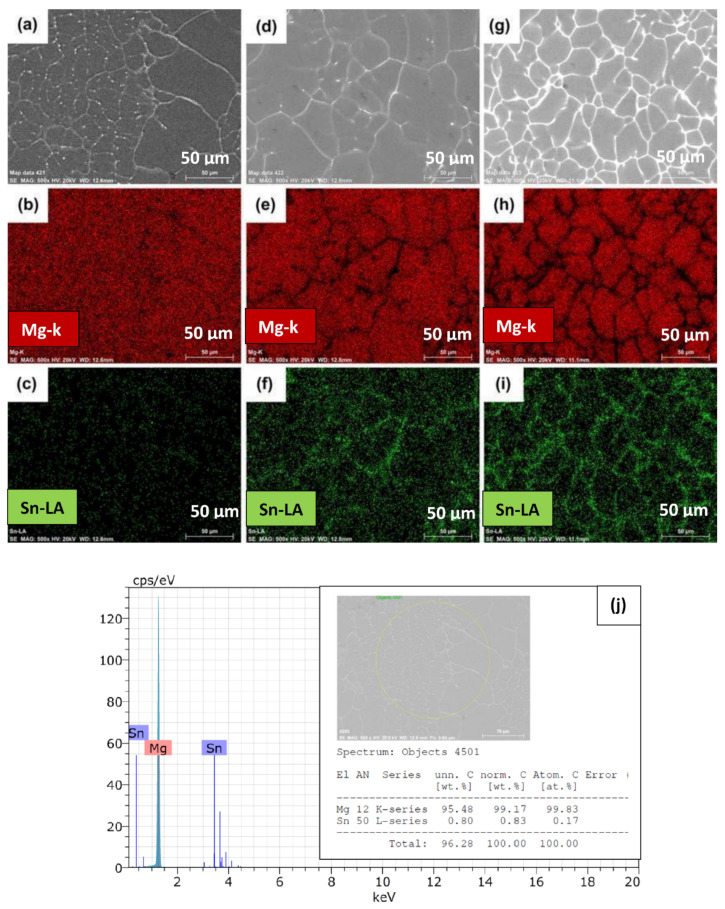
EDS elemental mapping of (**a**–**c**) Mg-1Sn; (**d**–**f**) Mg-5Sn; (**g**–**i**) Mg-10Sn alloys and EDS spectra of (**j**) Mg-1Sn; (**k**) Mg-5Sn; (**l**) Mg-10Sn.

**Figure 4 materials-15-06327-f004:**
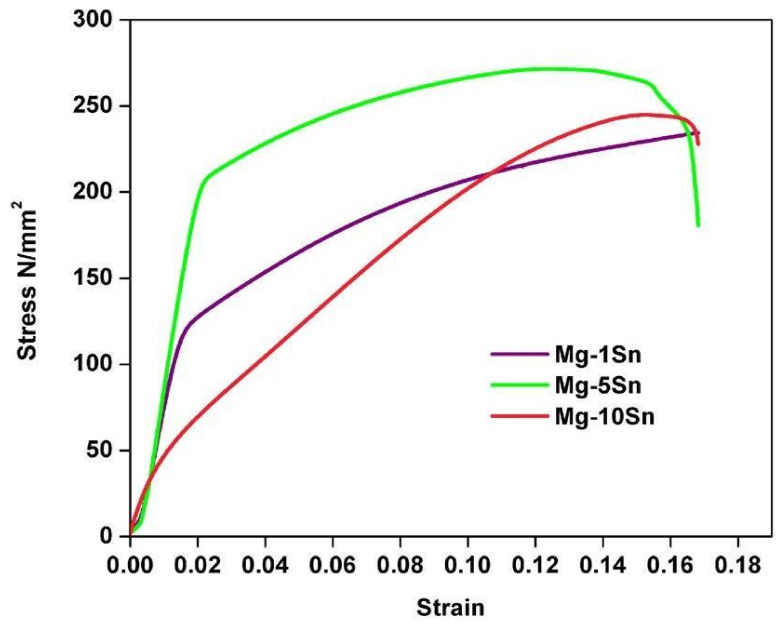
Compression curves of Mg-Sn alloys.

**Figure 5 materials-15-06327-f005:**
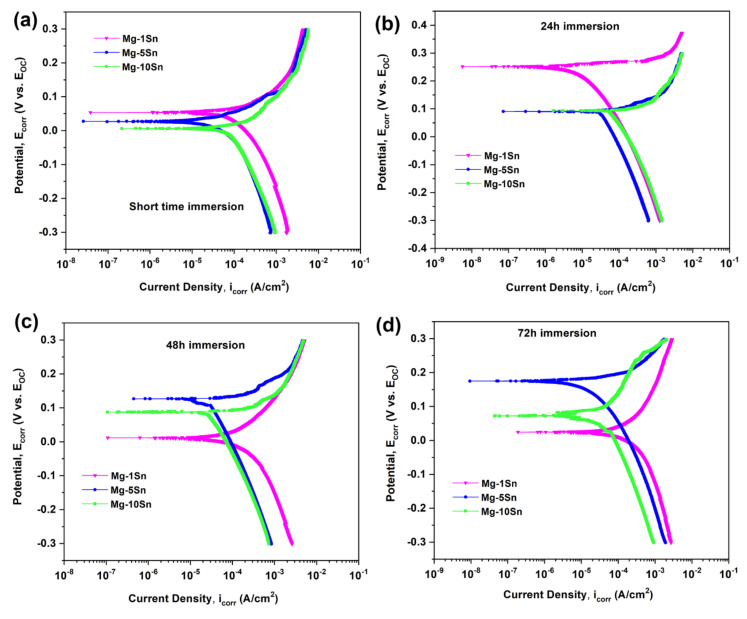
Tafel curves of Mg-Sn alloys under different immersion period. (**a**) ST (**b**) 24 h (**c**) 48 h (**d**) 72 h.

**Figure 6 materials-15-06327-f006:**
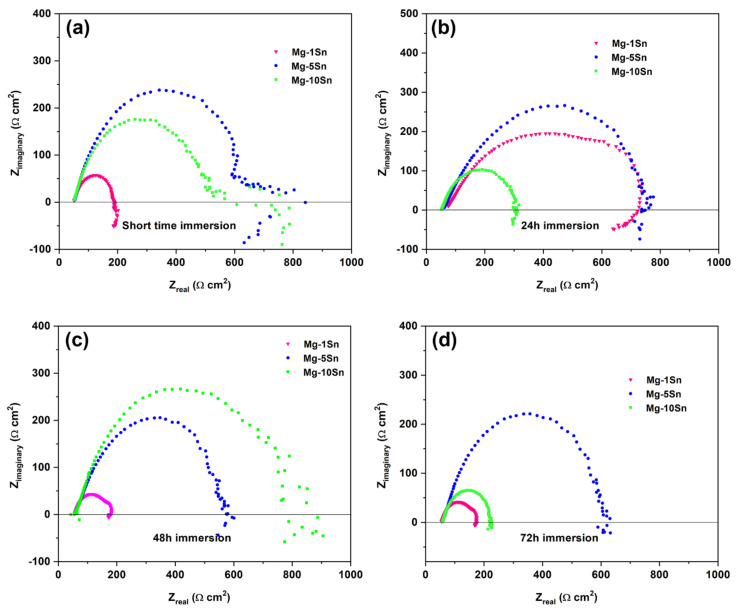
Nyquist curves obtained from EIS test on Mg-Sn alloys for different immersion period (**a**) Short term (**b**) 24 h (**c**) 48 h (**d**) 72 h.

**Figure 7 materials-15-06327-f007:**
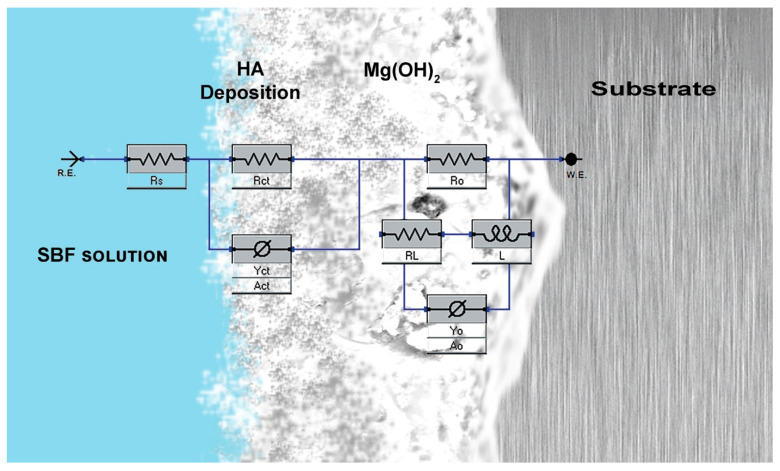
Equivalent circuit used for fitting the EIS curves.

**Figure 8 materials-15-06327-f008:**
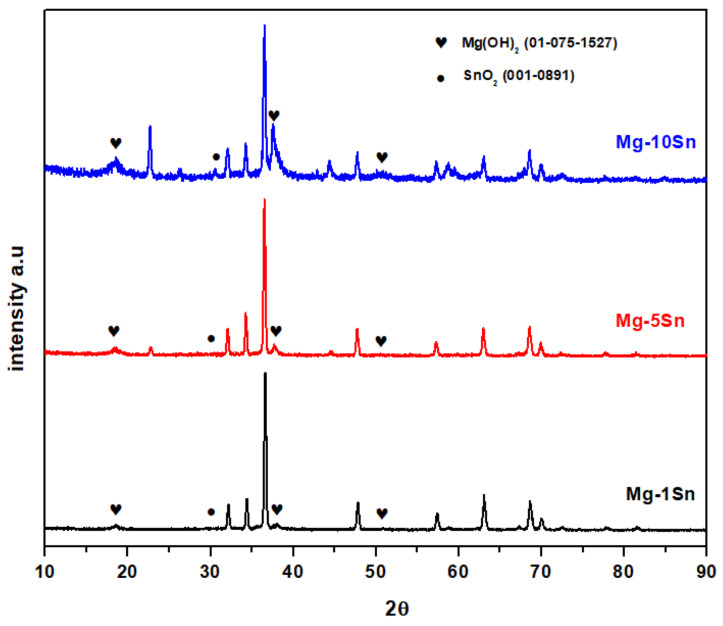
XRD spectra of Mg-Sn alloys after immersion test.

**Figure 9 materials-15-06327-f009:**
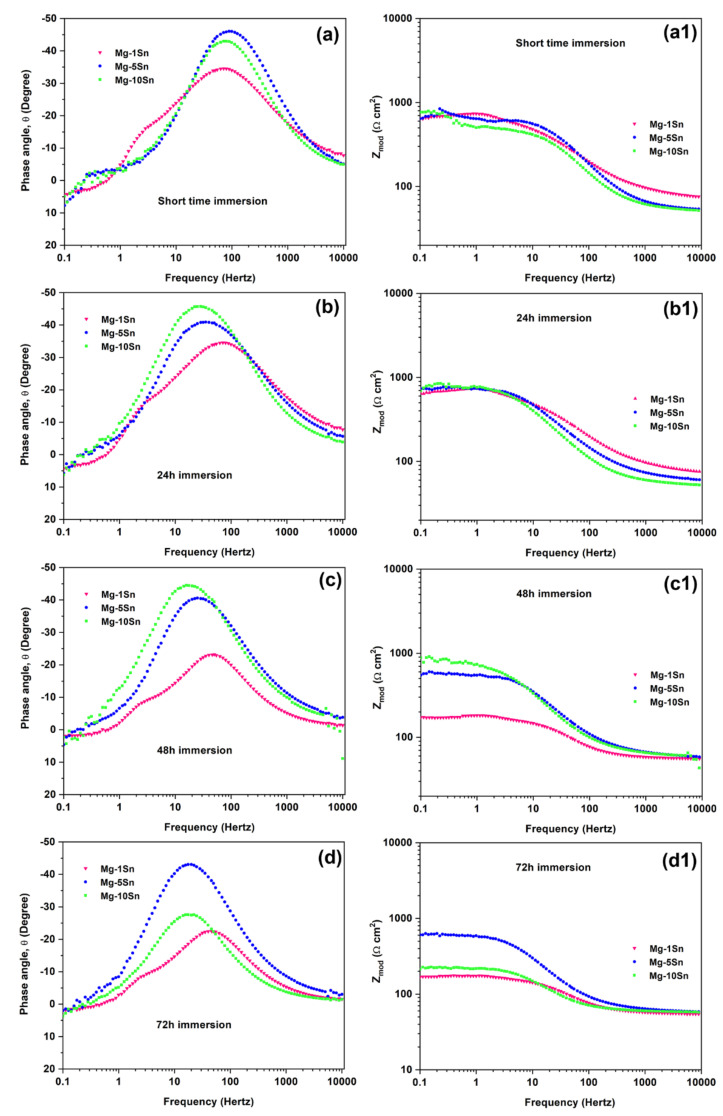
Bode curves (**a**–**d**) Phase angle (**a1**–**d1**) Impedance curve obtained from EIS test on Mg-Sn alloys for different immersion period (**a**,**a1**) Short term (**b**,**b1**) 24 h (**c**,**c1**) 48 h (**d**,**d1**) 72 h.

**Figure 10 materials-15-06327-f010:**
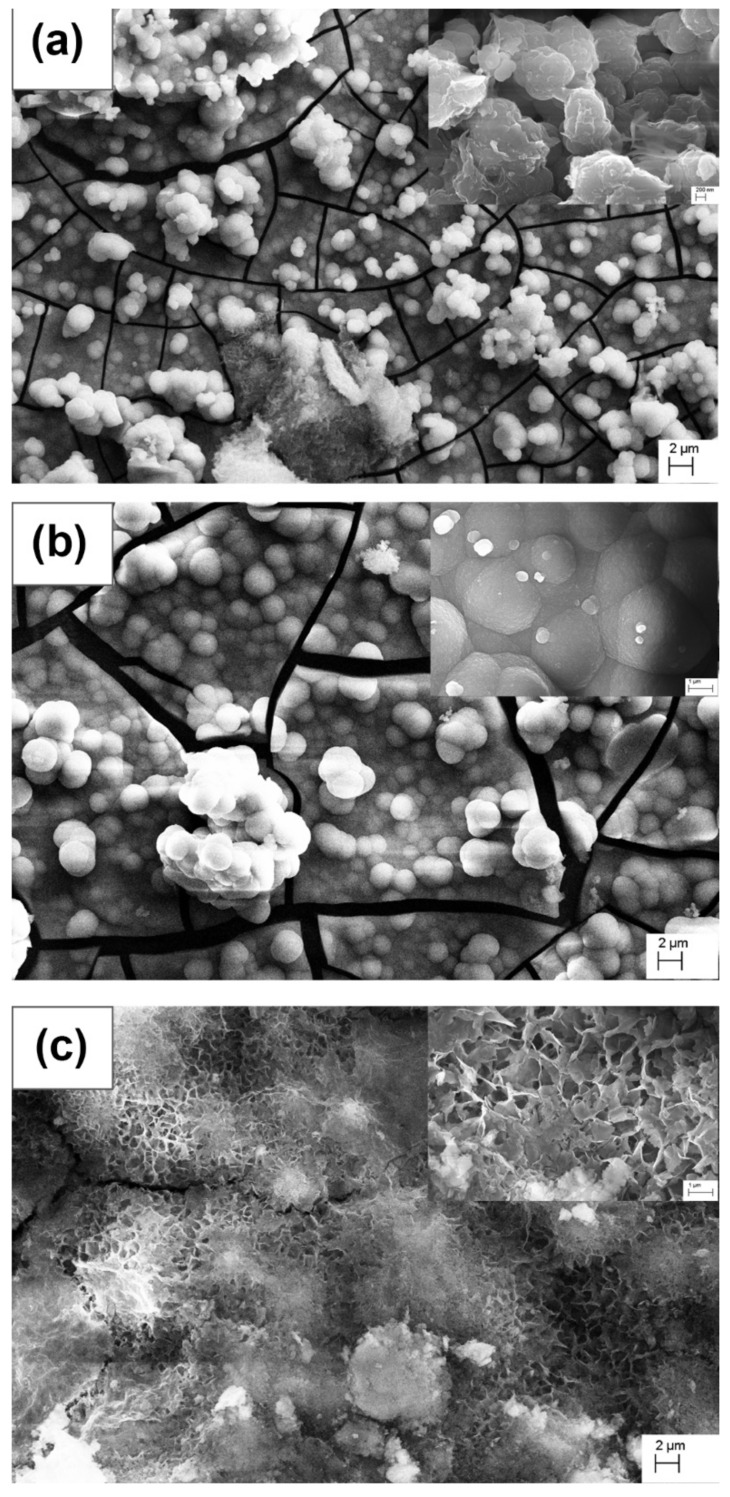
FESEM micrographs showing the bioactivity response of (**a**) Mg-1Sn; (**b**) Mg-5Sn; (**c**) Mg-10Sn alloys after 14 days of immersion.

**Figure 11 materials-15-06327-f011:**
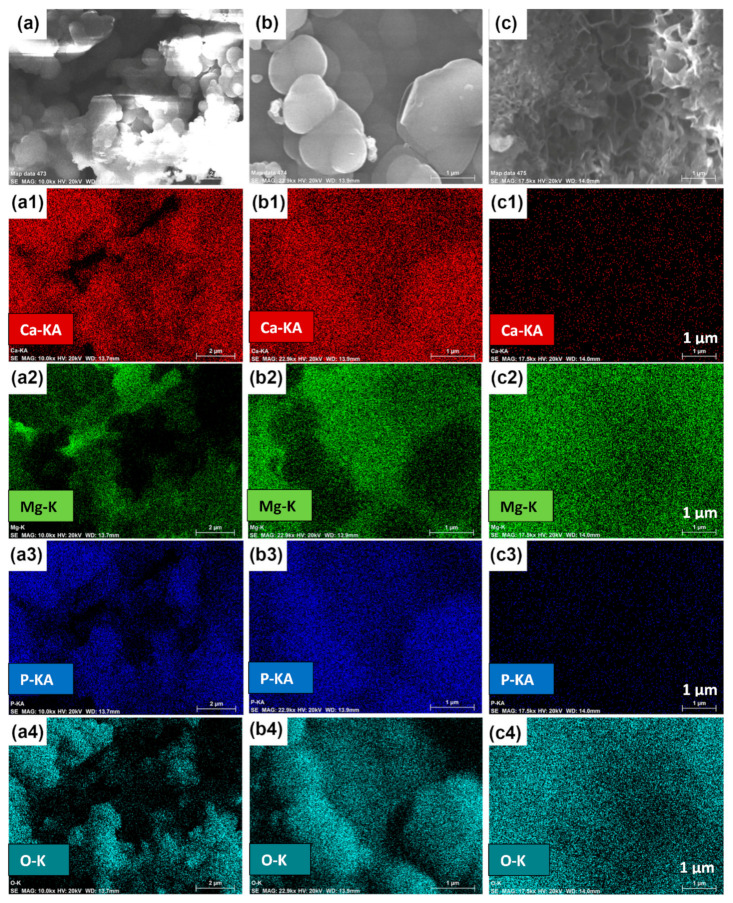
EDS elemental mapping showing the bioactivity response of (**a**–**a4**) Mg-1Sn; (**b**–**b4**) Mg-5Sn; (**c**–**c4**) Mg-10Sn alloys after 14 days of immersion.

**Table 1 materials-15-06327-t001:** Mechanical properties of Mg-Sn alloys.

Sample	UCS (N/mm^2^)	YS (N/mm^2^)	Elongation (%)	E (GPa)
Mg-1Sn	246.86 ± 6.3	134.34 ± 6.6	23.40 ± 2.1	12.80 ± 1.95
Mg-5Sn	271.39 ± 6.9	212.15 ± 6.2	12.22 ± 1.1	14.85 ± 1.6
Mg-10Sn	244.78 ± 5.7	90.41 ± 5.1	15.21 ± 1.6	4 ± 1.2

**Table 2 materials-15-06327-t002:** Corrosion results of Mg-Sn alloys estimated by Tafel extrapolation method.

Sample	i*_corr_* (×10^−6^ (A/cm^2^))				pH			
	ST	24 h	48 h	72 h	ST	24 h	48 h	72 h
Mg-1Sn	139 ± 1.24	36.9 ± 1.19	39.4 ± 0.98	37.2 ± 0.97	10.7	9.62	9.73	9.68
Mg-5Sn	38.3 ± 1.28	36 ± 2.05	27.9 ± 1.23	20.3 ± 0.75	9.71	9.6	8.51	8.34
Mg-10Sn	131 ± 1.63	49.5 ± 1.22	229 ± 1.24	449 ± 0.81	10.3	9.84	10.66	10.98

**Table 3 materials-15-06327-t003:** Corrosion rate measurement of Mg-Sn alloys from immersion test.

Sample	Open Circuit Potential (V)				Corrosion Rate (mm Per Year)			
	ST	24 h	48 h	72 h	ST	24 h	48 h	72 h
Mg-1Sn	−1.586	−1.594	−1.525	−1.548	0.056	0.14	0.20	0.18
Mg-5Sn	−1.562	−1.555	−1.554	−1.536	0.02	0.19	0.116	0.111
Mg-10Sn	−1.547	−1.513	−1.522	−1.494	0.04	0.09	0.21	0.20

**Table 4 materials-15-06327-t004:** Electrochemical data obtained by equivalent circuit fitting of EIS curves of Mg-Sn alloys.

Short Term Immersion									
Sample	L (H cm^2^)	R*_L_* (Ω.cm^2^)	R*_o_* (Ω.cm^2^)	Y*_o_* (S.sa/cm^2^ × 10^−4^)	A*_o_*	R*_s_* (Ω.cm^2^)	R*_ct_*(Ω.cm^2^)	Y*_ct_* (S.sa/cm^2^ × 10^−4^)	A*_ct_*
Mg-1Sn	235.8	0.118	421.3	48.9	0.661	52.22	453	0.375	0.832
Mg-5Sn	62.73	0.00411	230.9	95.1	0.614	54.23	582.8	0.245	0.847
Mg-10Sn	−0.543	10,300	14.4	1.77	0.699	49.96	123.9	0.275	0.945
**24 h Immersion**									
**Sample**	**L (H cm^2^)**	**R*_L_* (Ω.cm^2^)**	**R*_o_* (Ω.cm^2^)**	**Y*_o_* (S.sa/cm^2^ × 10^−4^)**	**A*_o_***	**R*_s_* (Ω.cm^2^)**	**R*_ct_*(Ω.cm^2^)**	**Y*_ct_* (S.sa/cm^2^ × 10^−4^)**	**A*_ct_***
Mg-1Sn	−2.641	721.4	36.83	0.573	0.664	51.9	217.5	1.04	0.931
Mg-5Sn	134.5	98,300	525.2	0.499	0.934	57.91	168.8	1.94	0.633
Mg-10Sn	647.7	1340	770.1	2.14	0.552	67.63	100.4	0.502	0.999
**48 h Immersion**									
**Sample**	**L (H cm^2^)**	**R*_L_* (Ω.cm^2^)**	**R*_o_* (Ω.cm^2^)**	**Y*_o_* (S.sa/cm^2^ × 10^−4^)**	**A*_o_***	**R*_s_* (Ω.cm^2^)**	**R*_ct_* (Ω.cm^2^)**	**Y*_ct_* (S.sa/cm^2^ × 10^−4^)**	**A*_ct_***
Mg-1Sn	1.415	279,000	18.92	3.75	0.632	55.97	735.3	0.915	0.835
Mg-5Sn	−106	188.8	124.2	5.17	0.572	56.47	438.3	0.653	0.915
Mg-10Sn	6.197	2.561	17.95	0.407	1.000	56.02	111.2	1.46	0.816
**72 h Immersion**									
**Sample**	**L (H cm^2^)**	**R*_L_* (Ω.cm^2^)**	**R*_o_* (Ω.cm^2^)**	**Y*_o_* (S.sa/cm^2^ × 10^−4^)**	**A*_o_***	**R*_s_* (Ω.cm^2^)**	**R*_ct_*(Ω.cm^2^)**	**Y*_ct_* (S.sa/cm^2^ × 10^−4^)**	**A*_ct_***
Mg-1Sn	913.1	0.276	135.3	0.181	0.999	54.04	164.1	3.90	0.733
Mg-5Sn	0.105	3220	52.34	9.22	0.549	56.6	502.9	0.823	0.908
Mg-10Sn	9.749	0.278	13.42	0.435	1	54.86	109.8	1.73	0.792

## Data Availability

Not applicable.

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
