# Peer review of "Insights of Microstructural Features and Their Effect on Degradation and the In Vitro Bioactivity Response of as-Cast Mg-Sn Alloys for Orthopedic Implant Applications"

_materials, 2022, doi:10.3390/ma15186327_

Round 1

Reviewer 1 Report

Notes on the article of Radha R and D. Sreekantha “Insights of microstructural features and their effect on degradation and in vitro bioactivity response of as-cast Mg-Sn alloys for orthopedic implant applications”

The paper reports results of studying of the effect of Sn content on the microstructure, mechanical properties, corrosion resistance and bioactivity in vitro response of as-cast Mg-Sn alloys. The authors report that mechanical properties, corrosion resistance and bioactivity of Mg-Sn alloys improved with increasing Sn content to 5 wt.% and decreased with the addition of 10 wt.% Sn. The authors have made many interesting tests, but since they only investigate the cast state of the alloys, the novelty of the article is somewhat doubtful. Despite this, it is an interesting and well-written report, which should be published after revisions that are listed below:

1)            What type of the standard did the authors use for immersion tests? According to ASTM_G1-03-E, cleaning samples in ethanol is not enough to completely remove corrosion products.

2)            The authors should give the values of the experimental error for each calculated value.

3)            It is not entirely clear why the authors call the apatite layer formation as bioactivity. Typically, bioactivity is an assessment of the effect of the alloy on cells. The authors should cite the studies that have used the same method of bioactivity assessment.

4)            The following typos need to be corrected:

P.4, L.112: «The mechanical properties» instead of «The mechanical proper- ties».

P.9: «then transforms to theMg2Sn.» instead of « then transforms to theMg2Sn.»

Author Response

Dear Reviewer,

Thank you so much for reviewing the manuscipt and gave suggestions for improving it for possible publication. Please see the attachment for the responses.

Reviewer 2 Report

Paper no. materials-1854545 titled Insights of microstructural features and their effects on degradation and in vitro bioactivity response of as-cast Mg-Sn alloys for orthopedic implant applications by Radha R and D Sreekanth

This is an interesting article in which the authors studied the microstructure and phase formation in binary Mg-xSn alloys (x=1, 5 and 10 wt.%).  The alloys were prepared by squeeze casting and their corrosion activity was studied in simulated body fluid (SBF) by electrochemical techniques. The best performance was found for the Mg-5Sn alloy. The paper is publishable subject to revision.

Major issues:

1.The XRD data (Fig. 1) should include Miller indices of the peaks and JCPDS file numbers of the identified phases.

2.You claim that three different phases were found in the alloys (Mg(ss), eutectic of Mg+Mg2Sn and Sn(ss), lines 175-177). The first two phases are easy to understand based on the Mg-Sn binary phase diagram. However, the existence of Sn(ss) is questionable. According to the equilibrium phase diagram of the Mg-Sn system, Sn can start form in the eutectic with Mg2Sn at a concentration higher than 30 wt. %. Sn. The 30 wt. % Sn is well above your concentration limits (1-10 wt. %). Furthermore, Sn has not been verified by the XRD (Fig. 1). You need to confirm this phase experimentally if you claim that it was present. Figs. 2 and 3 have a poor resolution. Could you, please, show a detail of the microstructure at magnification 2000x or higher? It is possible that the brightly lit areas were Mg2Sn (or the eutectic of Mg+Mg2Sn) and not Sn.

3.Do you have quantitive EDS point element analyses of the microstructure constituents (in wt.% or at. %)? If so, present them in the manuscript. The EDS maps are insufficient. They show only Mg-rich and Sn-rich areas which are already visible from back-scatter images. As such, they do not bring any new information.

4.Tafel curves (Fig. 5) are normalized to OCPs (open circuit potentials). However, the open circuit potentials of the materials are not given. You need to include these data in the manuscript. How did the OCPs of the alloys vary with time?

5.The calculated corrosion rates (CR) appear to be extremely high (0.7 – 8 mpy, i.e., 0.7 – 8 meters per year, see Table 3). Usually observed CR of Mg-Sn alloys is several millimeters per year in NaCl solution (2-4 mmpy), see, e.g. https://doi.org/10.1016/j.corsci.2019.108318 or https://doi.org/10.3390/ma15062025. Please, compare your CR results with previously published papers. Double check your calculation/dimension.

6.Corrosion experiments were conducted in an SBF solution (simulated body fluid). The SBF contains a NaCl solution as the major constituent. However, there are several commercially available SBF solutions. You need to specify the exact chemical composition of yours.

7.Mg alloys usually evolve a large amount of hydrogen when immersed in aqueous solutions. Have you studied/observed the H2 evolution? If so, the volumetric H2 results should be presented in the paper and used to calculate the corrosion rate. The volumetric CR can be directly compared with polarization results.

8.What were the corrosion products? You claim that Mg(OH)2 was formed (Fig. 7); however, this phase has not been verified. There are other corrosion products possible in addition to Mg(OH)2. Could you, please, show the XRD results of the post-corroded alloys? The EDS maps are insufficient, as they cannot unambiguously confirm the inorganic phases.

Technical points:

9.The authors should use an MDPI format thoroughly. The pages of the manuscript are not numbered. Furthermore, line numbers are absent from page 7 on. Please, number the pages and lines thoroughly to ease the orientation in the manuscript.

10.Increase the font size of the element denominations in EDS maps (Mg, Sn, Figs. 3 and 10) and size of the scale bars. They are hardly visible.

11.Highlights are not used in this journal.

Author Response

(The authors gave the same response as above.)

Round 2

Reviewer 2 Report

The authors answered most of my previous comments. The paper has been improved. However, few points still remain to be clarified:

1.The authors keep claiming that “segregated Sn” has been formed in the alloys (Fig. 2b). This phase has not been experimentally verified (see the XRD pattern in Fig. 1). It is highly unlikely that Sn was formed, based on the Mg-Sn binary phase diagram. Only Mg2Sn is possible. Therefore, you need to provide a solid experimental evidence, e.g., a high resolution TEM image coupled with an electron diffraction pattern, to confirm that Sn was really found in the alloys. An EDS analysis is insufficient (Fig. 3). The EDS data show some Sn concentration; however, it is impossible to distinguish between Sn and Mg2Sn by this technique alone. Therefore, a solid proof, e.g., an electron diffraction pattern, is necessary. Otherwise, you cannot claim it.

2.The authors clarified the dimension of the corrosion rate. However, ”Mils per year” (that is, milli-inch per year) is an imperial unit. It is not an SI unit of the corrosion rate. As such, the results cannot be directly compared with previous studies. Please, recalculate the corrosion rate into millimeters per year (1 mil = 0.0254 mm). Furthermore, compare your corrosion rates with previously published results of Mg-Sn alloys in NaCl solution (https://doi.org/10.1016/j.corsci.2019.108318, https://doi.org/10.3390/ma15062025) and make a note of it in the paper. If there is a difference of more than an order of magnitude, you should try to explain why.

Author Response

Dear Reviewer,

Kindly see the attachement for the responses to your valuable comments.
